# Computational Design of Novel Cyclic Peptides Endowed with Autophagy-Inhibiting Activity on Cancer Cell Lines

**DOI:** 10.3390/ijms25094622

**Published:** 2024-04-24

**Authors:** Marco Albani, Enrico Mario Alessandro Fassi, Roberta Manuela Moretti, Mariangela Garofalo, Marina Montagnani Marelli, Gabriella Roda, Jacopo Sgrignani, Andrea Cavalli, Giovanni Grazioso

**Affiliations:** 1Department of Pharmaceutical Sciences, Università degli Studi di Milano, Via L. Mangiagalli 25, 20133 Milano, Italy; marco.albani@unimi.it (M.A.); gabriella.roda@unimi.it (G.R.); 2Department of Pharmacological and Biomolecular Sciences, Università degli Studi di Milano, Via Balzaretti 9, 20133 Milano, Italy; roberta.moretti@unimi.it (R.M.M.); marina.marellimontagnani@unimi.it (M.M.M.); 3Department of Pharmaceutical and Pharmacological Sciences, Università di Padova, Via F. Marzolo 5, 35131 Padova, Italy; mariangela.garofalo@unipd.it; 4Institute for Research in Biomedicine (IRB), Via Chiesa 5, 6500 Bellinzona, Switzerland; jacopo.sgrignani@irb.usi.ch (J.S.); andrea.cavalli@irb.usi.ch (A.C.); 5Swiss Institute of Bioinformatics (SIB), University of Lausanne, Quartier UNIL-Sorge, Bâtiment Amphipôle, 1015 Lausanne, Switzerland

**Keywords:** peptide, LC3B, autophagy inhibitors, cancer, Atg8, LIR motif

## Abstract

(1) Autophagy plays a significant role in development and cell proliferation. This process is mainly accomplished by the LC3 protein, which, after maturation, builds the nascent autophagosomes. The inhibition of LC3 maturation results in the interference of autophagy activation. (2) In this study, starting from the structure of a known LC3B binder (LIR2-RavZ peptide), we identified new LC3B ligands by applying an in silico drug design strategy. The most promising peptides were synthesized, biophysically assayed, and biologically evaluated to ascertain their potential antiproliferative activity on five humans cell lines. (3) A cyclic peptide (named Pep6), endowed with high conformational stability (due to the presence of a disulfide bridge), displayed a K_d_ value on LC3B in the nanomolar range. Assays accomplished on PC3, MCF-7, and A549 cancer cell lines proved that Pep6 exhibited cytotoxic effects comparable to those of the peptide LIR2-RavZ, a reference LC3B ligand. Furthermore, it was ineffective on both normal prostatic epithelium PNT2 and autophagy-defective prostate cancer DU145 cells. (4) Pep6 can be considered a new autophagy inhibitor that can be employed as a pharmacological tool or even as a template for the rational design of new small molecules endowed with autophagy inhibitory activity.

## 1. Introduction

In living organisms, autophagy is a highly organized process that selectively captures proteins and old or damaged organelles using double-membrane vesicles called autophagosomes. When autophagosomes fuse with lysosomes, the contents within them are degraded by the acidic environment and lytic enzymes present in the lysosome [1,2]. The recycling ability of autophagy machinery is present in bacteria as well as in eukaryotic cells, allowing for the conservation of physiological conditions [2,3]. The autophagy machinery involves more than 50 proteins known as Atgs, but the ones responsible for forming the autophagosome and facilitating cellular trafficking are members of the Atg8 family. In mammals, Atg8 proteins (mAtg8) are categorized into two subfamilies: GABA-A receptor-associated protein (GABARAP) and microtubule-associated protein 1 light chain 3 (MAP1LC3), also known as LC3. The GABARAP subfamily consists of GABARAP, GARAPL1, and GABARAPL2, while the LC3 subfamily includes LC3A (with its two splicing variants LC3Aα and LC3Aβ, LC3B, LC3B2, and LC3C) [4]. Proteins within the same subfamily exhibit significant sequence similarities and perform similar functions within the cell. The GABARAP subfamily plays a crucial role in autophagosome closure and the recruitment of autophagy participants, while LC3 proteins primarily participate in the process of cargo recruitment [2]. Interference with autophagy by compounds capable of interfering with Atg8 proteins has still not been completely clarified, although the use of peptides or peptidomimetics capable of inhibiting Atg3–Atg8 interaction in Plasmodium falciparum has the potential to fight malaria [5,6]. In physiological conditions, autophagy degrades unfavorable components such as damaged organelles, pathogens, and oxidized biomolecules (proteins, DNA, and lipids) in response to oxidative stress, preventing cell damage. Conversely, in mammals, it has been demonstrated that dysregulations in the complex autophagy machinery are associated with various diseases, including neurodegenerative disorders [7], cardiomyopathies [8], infectious diseases [6,9], type II diabetes mellitus [10,11], hepatic steatosis [12], and cancer [13,14,15]. These pathologies are due to the autophagy corruption triggered by different stimuli originating from internal or external environmental factors. The role of autophagy in cancer is complex and depends on the phase and context of disease progression. It can play a pro-survival role, reducing cell death and promoting resistance to cytotoxic therapies, or it can be associated with cell death [16]. Numerous compounds have been identified to either stimulate or suppress autophagy, with the aim of obtaining therapeutic effects. These regulators serve as valuable research tools for investigating the intricate machinery of autophagy at a molecular level. Additionally, there is potential for their future development into promising drug candidates, with the aim of addressing cancer and other associated medical conditions [17,18,19,20]. The known autophagy regulators are active on biochemical pathways in which mTOR, class III PI3K (hVps34), Akt, V-ATPase, L-type Ca^++^ channel, Calpain, proteasome, tyrosine kinases, histone deacetylase, and some others are involved [17,18,19,20]. In our previous paper [18], we computationally designed two peptides (WC8 and WC10) endowed with high predicted and measured affinity on GABARAP, one of the subfamilies of the mAtg8 proteins. Interestingly, the treatments of human metastatic castration-resistant prostate cancer (CRPC) cells PC3 with WC8 and WC10 (from 1 to 10 µM) proved the high therapeutic potential of autophagy inhibition since the peptides were more active than paclitaxel, a common anticancer drug [21]. Nevertheless, LC3B is the most extensively studied Atg8 protein in humans due to its clear associations with cancer. In fact, a correlation between LC3B expression and higher tumor grade in clear cell renal carcinoma and other cancers has been established [22,23,24,25]. Moreover, studies on ovarian cancer cell lines revealed that the direct targeting of LC3B, to inhibit autophagy and promote apoptosis, enhances the sensitivity of cancer cells to chemotherapy [26]. Similarly, Quan et al. demonstrated that combination therapy with autophagy inhibitors and enzalutamide (a known antiandrogen for the treatment of prostate cancer) effectively induced bladder cancer apoptosis in vitro and in vivo [27]. Intriguingly, LC3 is initially expressed as pro-LC3 and undergoes cleavage by the cysteine protease Atg4B to form its cytosolic isoform LC3-I. Upon initiation of autophagy, LC3-I is bound to phosphatidylethanolamine (PE), becoming LC3-II, and localizes within the lipid membrane of developing autophagosomes [15]. Atg4B, together with other proteins involved in the autophagy machinery [28], and capable of interacting with LC3, possess a specific amino acidic sequence known as the “LC3 interacting region” (LIR), a small protein sequence containing four conserved residues. These can be briefly represented as a sequence of “X_0_–X_1_–X_2_–X_3_”, in which X_0_ is an aromatic residue (Trp/Phe/Tyr), X_1_ and X_2_ can be any amino acid (often acidic or hydrophobic residues), and X_3_ is a large hydrophobic residue like Leu, Val, or Ile [29]. For this reason, the LIR domain represents a wonderful starting point for designing ligands capable of interacting with the LC3 subfamily by applying the structure-based drug design approach. Among the proteins bearing the LIR domain, herein, we focus our attention on the proteome of Legionella pneumophila, an intracellular pathogen that produces a protein called RavZ. This has the peculiarity of triggering a decrease in the autophagy level of cells infected by *L. pneumophila*, limiting their ability to fight the infection through bacterial internalization in autophagosomes [30]. In this paper, we adopted a computational protocol that was successfully applied to select biologically active peptides [21,31,32]; we designed new peptides capable of binding LC3B, using as a template the structure of the bacterial protein RavZ. The newly designed RavZ analogs, also containing non-natural amino acids as well as conformational rigidification, were simulated through molecular dynamics (MD) simulations. Finally, the most promising peptides were synthesized, tested through biophysical experiments on recombinant LC3B protein, and in vitro assayed to demonstrate their biological effects on different cancer cell lines.

## 2. Results and Discussion

*Computational design of LIR2-RavZ analogs.* RavZ is an *L. pneumophila* protein capable of deconjugating LC3 proteins coupled to PE on autophagosomal membranes [33]. This is thanks to the presence of three LIR domains in the sequence. Among them, LIR2 (residues 27–32) seems to be the one responsible for the initial recognition of LC3, with it playing a crucial role in RavZ activity [33]. In fact, testing the binding of the LIR2-RavZ peptide (sequence DIDEFDLLEGDE, Table 1) on LC3B using isothermal titration calorimetry (ITC) and fluorescence polarization (FP), the measured dissociation constants (K_d_) were 360 nM and 550 nM, respectively, values comparable to that of the full RavZ in complex with LC3B (K_d_ = 260 nM) [33]. These data unambiguously demonstrate that the interaction between RavZ and LC3B can be mainly attributed to the LIR2 domain of the RavZ peptide (LIR2-RavZ, Table 1) [33]. Since structural data are not available in the protein data bank, in this study, we predicted the binding mode of LIR2-RavZ on LC3B by performing docking calculations and MD simulations on the resulting complex (see the Materials and Methods section for details). The visual inspection of the attained MD trajectory frames, together with the analysis of the ligand atoms’ root mean square fluctuation (RMSF) plot (see Figure 1A), suggested the high conformational mobility of the peptide *N*- and *C*-terms (mean RMSF value of 2.7 Å). Conversely, the residues located in the core of the peptide (residues 3–10) were almost stable (Figure 1A). The low fluctuation in the core was due to the presence of two salt bridges shaped by the side chains of LIR2-RavZ-Glu4 and -Asp6 with the ones of LC3B-Lys65 and -Arg69, respectively. Moreover, the hydrophobic tails of LIR2-RavZ-Phe5 and -Leu7 were inserted into a lipophilic pocket shaped by LC3B-Phe52, -Val54, -Pro55, -Val58, -Leu63 and -Ile66. Furthermore, the side chains of LIR2-RavZ-Glu9 and -Asp11 could create a H-bond network with LC3B-Lys30 and LC3B-Lys49, -Thr50, -Lys51 and -Thr50, respectively (Figure 1B).

Considering these data, we tried to design new LIR2-RavZ analogs with the aim of selecting new peptides endowed with increased affinity on LC3B, possibly by using unnatural amino acids to improve the metabolic resistance of the resulting peptides. In this process, three different strategies were applied:Rigidification: Designing cyclic peptides maintaining the original length of the peptide (12 residues) and inserting cysteines to create the disulfide bridge (replacing two residues, identified as having minor significance in the interaction with LC3B through alanine scanning);Terms protection through the amidation and acetylation of the *C-* and *N-* terms, respectively, to prevent peptide self-cyclization;Affinity maturation: Optimization of the sequence to attain new peptides with improved affinity on LC3B.

By applying these strategies, according to the “Materials and Methods” section, we initially calculated the binding free energy value (ΔG*) of the reference peptide LIR2-RavZ, attaining a value of −86.7 kcal/mol (SD = 13.1) (Table 1).

Then, with the aim of establishing which LIR2-RavZ residues were mainly involved in the interaction with LC3B, and recognizing the hot and non-hot spots, computational alanine scanning was carried out (Appendix A). From these calculations, it could be seen that all residues seemed critical for interaction with the target since their substitution by alanine led to an increase in the estimated ΔG* values. Thus, in the first attempt to design new potent LIR2-RavZ analogs, we tried to limit the conformational flexibility of the peptide by creating a disulfide bridge between the residues in positions 10 and 12. Position 12 was chosen because the Ala substitution of the residues in position 12 led to peptides with estimated ΔAffinity not greatly diverse from the parent peptide, while a residue without a side chain (a Gly) is naturally present in the sequence. The resulting peptide, Pep1, was simulated in complex with LC3B through MD simulations and the further application of the molecular mechanics-generalized Born surface area (MM-GBSA) approach suggested that it possessed a ΔG* value of −87.5 kcal/mol (SD = 8.7) (Table 1). This value, effectively equal to the one calculated for the natural peptide LIR2-RavZ, suggested that the presence of conformational rigidification at the C-term did not elicit any change in the LC3B/peptide interaction strength. With variance, the Cα atoms’ RMSF, evaluated by MD simulations, demonstrated that the cyclic peptide Pep1 fluctuated around a value lower than the one displayed in the MD simulations on the parent peptide, suggesting that the conformational rigidification stabilizes the new peptide on the LC3B surface (Appendix A). In the second attempt, the peptide termini were protected by amidation and acetylation, with the aim of reducing the *N-*/*C-* reciprocal interaction and the consequent creation of a cyclic peptide in solution. Thus, performing MD simulations and MM-GBSA calculations again on the termini-protected peptide (Pep2), the predicted ΔG* value of Pep2 decreased to the value of −109.1 kcal/mol (SD = 7.2), a value almost 21 kcal/mol lower than the unprotected peptide (Table 1). Subsequently, the “affinity maturation” procedure was applied to Pep2, with the aim of designing new Pep2 analogs with primary structure endowed with improved complementarity with LC3B. This approach allowed for the identification of new peptides through the substitution of the non-hot spot residues with new ones displaying higher affinity to the LC3B binding site. Specifically, to identify the non-hot spots, a new alanine scanning calculation was accomplished on Pep2, and, at the end of calculations, the attained results suggested that the non-hot spot residues were the ones in positions 1, 2, 3, and 6 (Appendix A). Consequently, in the next affinity maturation calculation, Pep2 was randomly mutated by new amino acids, creating a final library of 100,000 peptides, containing *l*- and *d-*, side chain-protected, phosphorylated, and unnatural amino acids. The newly designed peptides were ranked considering the predicted affinity on the target and the resulting three top-ranked peptides (Pep3–5) were simulated in complex with LC3B through MD simulations, with their ΔG* values finally being calculated via the MM-GBSA approach as well. Interestingly, the estimated ΔG* values of these peptides were almost 30 kcal/mol lower compared to the parent peptide Pep2, confirming the strength of the applied affinity maturation procedure (Table 2).

Among them, Pep3 was the most promising one since it showed a ΔG* value of −149.8 kcal/mol. The sequence of this peptide contains two phosphorylated residues, in positions 6 and 7, together with two benzyl (Bz)-protecting groups on the side chains of Glu2 and Thr3. The Cα RMSF plot retrieved from MD simulations suggested that Pep3 was firmly bound on the LC3B surface, displaying a mean RMSF value of 0.9 Å (Figure 2A). Interestingly, the *N*- and *C*-terms displayed low fluctuation, suggesting that they could create anchoring contacts with LC3B. Indeed, the acetyl group protecting the Pep3 N-terminus formed a water-mediated hydrogen bond with LC3B-Asn59, while the side chain of the same residue established a salt bridge with the one of LC3B-Lys65. The Bz group attached to the side chain of Pep3-Glu2 reached LC3B-His57, resulting in a π-π interaction. Pep3-Glu4′s side chain formed a salt bridge with LC3B-Arg69, while Pep3-Phe5′s side chain was inserted into the hydrophobic pocket defined by LC3B-Ile35, -Val54, -Pro55, -Leu63, and -Ile66. Thr(P) in position 5 of Pep3 was positioned in proximity to the positively charged area formed by the side chain of LC3B-Lys30, whereas the Tyr(P) in position 7 interacted with the alkaline moieties of LC3B-Lys49, -Thr50, and -Lys51. The latter LC3B residues established a salt bridge with the side chain of Pep3-Glu9 while the disulfide bridge (Cys10-Cys12) of Pep3 was inserted into the crevice sized by LC3B-Leu53, -Phe108, -Ile34, and -Pro32. The amidated C-terminus of Pep3-Cys12 created an H-bond network with LC3B-Asp19 and -Lys51. All of these contacts (Figure 2B) strongly stabilize Pep3 on the LC3B LIR binding domain, as displayed by the RMSF plot (Figure 2A).

*Biological activity of Pep3.* Pep3, which proved to be the most promising peptide among the ones investigated, was acquired by Pepmic (Pepmic Co., Ltd., Suzhou, Jiangsu, RPC), and it was assayed using preliminary cell viability assays on the PC3 prostatic cancer cell line (Figure 3). PC3 cells were treated with a range of doses from 0.0025 µM to 5 µM for 24, 48, and 72 h, and viability assays were performed. The results obtained highlight that Pep3 exerted a significant cytotoxic effect at both 2.5 and 5 µM doses after 24 h of treatment, while at 48 h, only 5 µM was effective. Furthermore, the treatment for 72 h did not modify cell viability compared to the control cells, resulting in a lack of efficacy (Figure 3A). We then analyzed the effect of Pep3 on LC3 expression. The analysis of the ratio of LC3-II/LC3-I was not different at any time considered, but in the treated samples, there was a reduction in both LC3-I and LC3-II expression compared to the control sample after 24 h and to a lesser extent after 48 h. It is possible to speculate that in the presence of basal physiological autophagy activation, Pep3 binds on the LC3 precursor, preventing the formation of both LC3-I and lipidated LC3-II (Figure 3B). The same effect on LC3-I and LC3-II expression was obtained with the treatment of PC3 cells with the autophagy inhibitor 3-metyladenine (3-MA, 1 mM) (Sigma-Aldrich), which blocks the formation of phagophore, inhibiting the phosphoinositide 3-kinases (PI3K) [34]. It is possible that the effectiveness of Pep3 decreased in the considered time from 24 h to 72 h, possibly due to the peptide’s low metabolic stability.

Consequently, we surmised that a new peptide containing D-amino acids in its sequence (Pep6, Table 2) could be more resistant to peptidase activity. Therefore, we used the peptidecutter web service “https://web.expasy.org/peptide_cutter (accessed on 8 January 2024)” to predict the Pep3 cleavage sites. Here, by inserting the Pep3 sequence (without any side chain derivatization) and selecting the prediction of cleavage sites cleaved using proteases (excluding the bacterial ones and chemical reagents), it appeared that the residues in the middle positions could be the most susceptible. Consequently, we designed a Pep3 analog (Pep6) in which the residues at positions 3, 5, 6, 10, and 11 were mutated by D-amino acids. The predicted ΔG* value of the resulting peptide (Pep6) was −127.1 kcal/mol (SD = 10.8), a value about 23 kcal/mol higher than the one of Pep3 but still about 40 kcal/mol lower than the ΔG* value calculated for the template peptide LIR2-RavZ. The MD simulations on the LC3B/Pep6 complex suggested that the overall conformational stability of the peptide on the LC3B remained essentially like that of Pep3 (0.9 Å, Figure 2A), since the Pep6 mean Cα RMSF value was 0.7 Å (Figure 4A). Moreover, the “simulation interaction analysis” tool, together with the visual inspection of the MD trajectory frames, suggested that the alkaline side chains of LC3B-Lys65, -Arg69, -Arg70, and -Lys49 could create H-bond assisted salt bridges with Pep6-Asp1, -Glu4, -Tyr(P)7, -Glu9, and -Asp11 (Figure 4B). Moreover, Pep6-Asp1 established H-bond interactions with LC3B-Ser61 and -Glu62. Both the amide and carbonyl backbone atoms of Pep6-Leu8 create an H-bond network with the backbone atoms of LC3B-Leu53, while its lipophilic chain was in contact with an LC3B pocket shaped by His27 and Leu53. The disulfide bridge remained in contact with LC3B-Leu53, -Phe108, -Ile34, and -Pro32. The side chain of Pep6-D-Tyr(P) created a salt bridge with LC3B-Lys30, and the phenyl ring of Pep6-D-Phe5 was in contact with the imidazole ring of LC3B-His57, creating a π-π interaction (Figure 4B). Finally, the amidated C-terminus of Pep3-D-Cys12 created an H-bond network with LC3B-Asp19 and -Gln26. For a comparison with a known LC3 binder, in 2021, Fan et al. reported on covalent ligands capable of reducing autophagic structure formation and subsequent substrate degradation [35]. Superimposing our LC3B/Pep6 complex with LC3B covalently bound with compound a4 at Lys49 (PDB accession code 7ELG), we noted that the small molecule is aligned on the Pep6 residues in position 7–9, amino acids T(P)Y(P)L (Appendix A). Interestingly, both Lys49 and Lys51 are located in the LIR docking site, and their occupation by the presence of any ligand impairs the protein–protein interaction between LC3B and its biological partners involved in autophagy. Thus, we can speculate that the presence of our peptide in the LIR docking site could avoid pro-LC3B maturation through prodomain removal catalyzed by Atg4. Additionally, the presence of the peptide in proximity to Lys49 and Lys51 could prevent endogenous acetylation and deacetylation modifications in cells [36]. In fact, it has been reported that the acetylation of these residues blocks the interactions of LC3B with p62 and Atg7 proteins, leading to the accumulation of LC3B-I and autophagy inhibition [37,38]. Nevertheless, additional biochemical experiments are needed to better understand the action mechanism of our peptides.

Pep6 was then acquired by Pepmic (Pepmic Co., Ltd., Suzhou, China), and it was tested using cell viability assays on a non-cancerous cell line (PNT2, to evaluate the cytotoxicity) and PC3, DU145 CRPC cells, A549, and MCF-7 cancer cell lines. These cell lines were chosen since it has been demonstrated that their growth is influenced by autophagy inhibition [23,24,25].

*Biological activity of LIR2-RavZ and Pep6 on cancer cell lines.* The biological activity of LIR2-RavZ and Pep6 (from 0.0025 to 5 µM) was evaluated with MTT cell viability assay on PC3 cells (Figure 5A). Treatment for 72 h with both compounds determines a reduction in cell viability with greater efficacy of Pep6 compared to LIR2-RavZ at all doses used (Figure 5A). The effects of both compounds were then evaluated on the CRPC cell line DU145, which is defective in autophagy due to its lack of functional ATG5 [39]. Treatment with LIR2-RavZ and Pep6 (5 µM for 72 h) in DU145 cells was ineffective, confirming that the antitumoral activity of the peptides was attributable to autophagy activation (Figure 5B). We then analyzed the expression of LC3 and SQSTM1/p62 (p62) in PC3 cells treated with both compounds for 48 h. In analogy to results obtained with Pep3, both LIR2-RavZ and Pep6 significantly reduced the expression of LC3-I and LC3-II without changing the LC3-II/LC-I ratio. Furthermore, p62 expression was significantly increased after treatment with both compounds (Figure 5C). We also evaluated the ability of Pep6 to interfere with trehalose-dependent autophagy activation. PC3 cells were co-treated with Pep6 (5 µM dose for 48 h) and trehalose (100 mM for 48 h) (Sigma-Aldrich). The Western blot (WB) shown in Figure 5D highlights that Pep6 in association with trehalose was able to decrease the LC3-II/LC3- ratio compared to trehalose alone. These data suggest that Pep6 inhibits trehalose-induced autophagy, causing less autophagosome formation. We can conclude that the antitumoral effect of Pep6 in PC3 cells is more stable over time than Pep3, and this antitumoral activity is related to autophagy inhibition.

The biological activity of different concentrations of LIR2-RavZ and Pep6 (from 0.0025 to 5 µM) was evaluated with an MTS cell viability assay on non-cancerous PNT2 prostate cells and different cancer cell lines such as A549 and MCF-7 (Figure 6). The results reported in Figure 6A show that 96 h post-treatment, none of the tested samples displayed significant cytotoxicity at the concentration of 5 mM (cell availability > 90%), confirming the excellent biocompatibility and potential pharmacological selectivity for tumor cells. Indeed, as shown in Figure 6B,C, a reduction in cell viability (expressed as percent-age % of viable cells) was observed in A549 and MCF-7 cells treated with LIR2-RavZ and Pep6 compared to the untreated control cells. These in vitro data demonstrate that the compounds exhibited slight anticancer activity, with a reduction in cell viability especially at the highest tested concentration (Figure 5A and Figure 6B,C). More in depth, the obtained results indicated that A549 cells are more responsive than MCF-7 cells. In fact, the cell viability for A549 was 81.1% and 76.7% for LIR2-RavZ and Pep6, respectively, whereas for MCF-7 cells, the viability was 85.4% and 80.7% for LIR2-RavZ and Pep6, respectively. Finally, it is possible to speculate that, since these peptides did not elicit any toxicity on PNT2 cells, they could have an interesting application for the inhibition of pro-survival autophagy induced by chemotherapy or anti-tumor drugs, reducing the onset of drug resistance [26].

*Biophysical assays.* To finally confirm that Pep6 can really affect autophagy machinery by binding on LC3B, MST experiments were performed on human recombinant His-tagged LC3B protein [21]. This biophysical technique quantifies the interactions between two entities in contact in the liquid phase, avoiding any sample immobilization, as needed in other approaches, such as in the case of surface plasmon resonance (SPR) [40]. Initially, the dissociation constant (K_d_) value of the LIR2-RavZ peptide (as a positive control) was estimated through MST experiments in order to validate the applied biophysical method, leading to a K_d_ value of 428 ± 162 nM (Figure 7A). This value resulted in being comparable with the one reported by Yang A. and co-workers, where LIR2-RavZ was tested through ITC and FP, displaying dissociation constants of 360 nM and 550 nM, respectively [33]. Finally, we evaluated the binding of Pep6 to the LC3B protein using MST, obtaining a K_d_ value of 159 ± 56 nM (Figure 7B), which is a value about three times lower compared to the one acquired for LIR2-RavZ. The experimental conditions of the MST experiments carried out are detailed in the Material and Methods section.

## 3. Materials and Methods

*Computational studies.* The LC3B computational model utilized in this study was constructed using the 3D coordinates of the chain A of the LC3B/FYCO1-LIR complex, retrieved from the Protein Data Bank (PDB accession code 5D94) [41]. The structure available in this databank represents the sequence from amino acids 5 to 123. During the maturation process, the signal peptide spanning amino acids 121 to 125 is cleaved from the C-terminal end. Therefore, the X-ray, as well as the computational models built using it, represents the pro-LC3B state of the LC3 maturation process. In this paper, we named it “LC3B” for simplicity. The LC3B model underwent optimization using the “Protein Preparation Wizard” tool within Maestro Software (release 2021-2, Schrödinger, LLC, New York, NY, USA). This tool facilitated the initial steps of system setup, encompassing: (1) assessing the protonation states of residues at pH 7.4, (2) verifying residue completeness, (3) resolving atomic clashes, and (4) applying the OPLS4 force field [42]. The docking of LIR2-RavZ was executed using the “Peptide Docking” tool in Maestro software (release 2021-2, Schrödinger, LLC, New York, NY, USA). This process involved defining a grid, creating a centroid through the “Centroid of Selected Residues” option, and selecting the residues in complex with the LIR domain in the X-ray structure, such as Arg10, Lys30, Tyr50, Lys51, Leu53, Arg69, and Phe108. The grid dimensions accommodated a linear 12-residue peptide. The sequence of LIR2-RavZ (Table 1) was introduced, cis amide bonds were deactivated, 150 poses were generated, and the Glide score [43] served as the scoring function, resulting in the LC3B/LIR2-RavZ starting complex. After this, alanine scanning was accomplished using the “Residue Scanning” tool available in Maestro (release 2021-2, Schrödinger, LLC, New York, NY, USA). Each peptide residue was individually mutated to alanine to pinpoint crucial hot/non-hot spots for LC3B and LIR2-RavZ interaction. The same tool was employed for the affinity maturation process. All residues, except the two cysteines, went through simultaneous mutations including natural and unnatural residues, selected from the library available in the tool. Affinity maturation utilized Monte Carlo optimization with a maximum of 100,000 steps, focusing on optimizing affinity and generating a maximum of 150 structures. Subsequently, a cubic box of water molecules, represented by the TIP3P model, was built around the protein–ligand complex, and subsequent system energy minimization was followed by 250 ns long MD simulations, using the Desmond algorithm in Maestro (release 2021-2, Schrödinger, LLC, New York, NY, USA). The “Simulation Interactions Diagram” tool evaluated peptide and ligand stability (see Appendix A, for the Cα atom RMSD plots of the main simulated systems). Finally, ligand binding free energy (ΔG) calculations were conducted using the Prime algorithm within Maestro software (release 2021-2, Schrödinger, LLC, New York, NY, USA), employing the MM-GBSA algorithm. The single-trajectory approach was adopted, neglecting entropy contributions to the binding free energy. The resulting binding free-energy values were denoted as ∆G* by our group [21,44,45]. This protocol was applied to all peptides under investigation (Table 1 and Table 2).

*Cell lines*. The PNT2 (European Collection of Authenticated Cell Cultures, ECACC, UK) human prostate cell line was purchased from Sigma-Aldrich (St. Louis, MO, USA). PC3 and DU145 human CRPC cell lines, A549 human lung cancer cell lines, and MCF-7 human breast cancer cell lines were purchased from the American Type Culture Collection (ATCC, USA). PNT2 cells were cultured at 37 °C and 5% CO_2_ in RPMI 1640 (Gibco Laboratories) supplemented with 10% FBS (Gibco Laboratories), 1% 100 µ/mL penicillin/streptomycin (Gibco Laboratories), and 2% L-glutamine (Gibco Laboratories). PC3 and DU145 cells were maintained in RPMI 1640 medium (EuroClone, Milano, Italy) supplemented with 7.5% (PC3) and 5% (DU145) FBS (Gibco Laboratories), 1% L-glutamine, and antibiotics (100 IU/mL penicillin G). A549 and MCF-7 cells were cultured in Dulbecco’s modified Eagle medium (DMEM, Lonza, Switzerland) supplemented with 10% fetal bovine serum (FBS, Gibco Laboratories, USA), 1% 100 u/mL penicillin/streptomycin (Gibco Laboratories), and 1% L-glutamine (Gibco Laboratories).

*Cytotoxicity studies*. PNT2, A549, and MCF-7 cells were seeded at a density of 1 × 10^4^ cells/well in 96-well plates and maintained under standard growth conditions. After 24 h, the cells were treated for 1h at 0.0025 μM, 0.025 μM, 2.5 μM, and 5 μM to a final volume of 100 μL. After 96 h, cell viability was assessed using MTS assay, according to the manufacturer’s protocol (Cell Titer 96 Aqueous One Solution Cell Proliferation Assay; Promega, Nacka, Sweden) using a 96-well-plate spectrophotometer (Varioskan Flash Multimode Reader; ThermoFisher Scientific, Waltham, MA, USA) set at λ = 490 nm. The absorbance value of untreated cells was set at 100% (control), and the viability of treated cells was expressed as a percentage of the control. PC3 and DU145 cells were plated at a density of 3 × 10^4^ cells/well in 24-well plates. After 48 h, the cells were treated with compounds at 0.0025 μM, 0.025 μM, 2.5 μM, and 5 μM doses. After 24 h, 48 h, or 72 h, cell viability was analyzed using 3-(4,5-dimethylthiazole-2-yl)-2,5-diphenyltetrazolium bromide (MTT) (Sigma-Aldrich, St. Louis, MO, USA) assay. At the end of the treatment, the medium was replaced with MTT solution (0.5 mg/mL) in RPMI without phenol red and FBS. After 30–45 min at 37 °C, the precipitate of formazan was dissolved with isopropanol. Absorbance (λ = 550 nm) was measured through the use of an EnSpire Multimode Plate reader (Perkin Elmer, Milano, Italy). Three independent experiments were performed for each condition. 

*Western blot (WB) assay*. To investigate the effects of compounds on autophagy modulation, PC3 cells were plated at 2 × 10^5^ cells/dish in 6 cm dishes for 24 and 48 h. The cells were then treated and adherent, and floating cells were harvested and lysed in RIPA buffer. Protein extracts (10–20 μg) were resuspended in reducing sample buffer (Bio-Rad Laboratories, Segrate, Milano, Italy) and heated at 95 °C for 5 min. Following electrophoretic separation via SDS-PAGE, the proteins were transferred onto PVDF or nitrocellulose membranes. After blocking, the membranes were incubated with anti-LC3 (L8918) (Sigma-Aldrich, St. Louis, MO, USA)) and anti-SQSTM1/p62 (PA5-20839) (Thermo Fisher Scientific, Watham, MA, USA) primary antibody. Peroxidase-conjugated secondary anti-rabbit or anti-mouse antibodies were used for 1 h at room temperature, and membranes were processed using the enhanced chemiluminescence kit Cyanagen Ultra (Cyanagen, Bologna, Italy). In each WB experiment, alpha-tubulin expression (T6199) (Sigma-Aldrich) was evaluated as a loading control. 

*MST Experiments*. The binding of the peptides on LC3B protein was assessed by employing the Monolith NT.115^Pico^ instrument (NanoTemper Technologies GmbH, München, Germany), which allows for estimation of the dissociation constant (K_d_) comprising the concentration range from 1 pM to mM. Specifically, His-tagged human recombinant LC3B (catalog number 14555-H07E, Sino Biological, Beijing, China) was red-labeled using the dedicated His-Tag Labeling Kit RED-tris-NTA 2nd Generation from NanoTemper Technologies (Product No. MO-L018) for 30 min at room temperature. The “Binding Affinity” mode of the dedicated software MO.Control v1.6 (NanoTemper Technologies GmbH, München, Germany) was used to perform the MST experiments. In detail, a fixed concentration of red-labeled LC3B (10 nM) was mixed with sixteen 1:1 serial dilutions of LIR2-RavZ peptide (concentration range: 15.6 µM–0.477 nM), which was used as a positive control, and Pep6 (concentration range: 31.3 µM–0.954 nM), respectively. Both interacting species were dissolved in PBS-T buffer (phosphate-buffered saline + 0.05% Tween™ 20) from NanoTemper Technologies with 2.5% dimethyl sulfoxide (DMSO) for molecular biology (Product No. D8418; Sigma-Aldrich, Saint Louis, USA). The samples were incubated for 15 min at room temperature, then filled into standard capillaries (Product No. MO-K022; NanoTemper Technologies GmbH, München, Germany), and, finally, measured through the employment of an excitation power of 20% and the medium MST power (40%) fixing the temperature at 25 °C. The auto-fluorescence of each peptide was evaluated before proceeding to the determination of the K_d_. The experimental data were processed by employing dedicated MO.Affinity Analysis software v2.3 (NanoTemper Technologies GmbH, München, Germany) and the K_d_ model for fitting the binding curve, while the figures were generated using GraphPad Prism software v8.0.2 (GraphPad, Boston, MA, USA). The MST analysis report can be consulted in the Appendix A, namely Appendix A (dataset overview) and Appendix A (MST traces and capillary scan graphs).

## 4. Conclusions

Our computational approach aimed at designing new peptides endowed with high affinity on a specific target led to the desired goal. In fact, in this study, starting from the sequence of a known peptide demonstrating affinity on LC3B, we applied an approach combining MD simulations and MM-GBSA calculations to computationally predict the binding affinity of new peptides obtained by mutating the sequence of a known peptide (affinity maturation process). As can be seen in Figure 1, Figure 2 and Figure 4 and Table 1 and Table 2, using our approach, both the stability and the affinity of the new peptides were significantly improved. In fact, considering the predicted binding modes of LIR2-RavZ and the ones of Pep3 and Pep6, the new peptides can create numerous electrostatic interactions with the LC3B positive charged area shaped by Lys30, Lys65, Arg69, and Arg70 residues, explaining the low predicted ΔG* values. Moreover, the backbone rigidification conferred by the presence of the disulfide bridge in the sequences led to a low fluctuation in the peptides on the LIR binding site of LC3B, together with better filling of the LC3B basin shaped by Ile23, Leu58, and Phe108 residues (close to the W-site). Finally, it has to be stressed that Pep6 contains D-amino acids and has side chain modification to improve metabolic resistance. We are aware that cellular phosphatases could remove the phosphate groups on serine, threonine, and tyrosine. Therefore, we understand that the phospho-peptides Pep3 and Pep6 may be subject to hydrolysis in the cellular environment; in particular, the L-Tyr(P) at position 7 may be the most susceptible. Nevertheless, we anticipate that this susceptibility to cellular phosphatases could be reduced by incorporating the disulfide bridge, the Bz groups, and the D-amino acids in the sequence.

Thus, based on the biological assessments mentioned earlier, it is reasonable to hypothesize that, if adequately delivered into the intracellular compartment via nanocarriers or liposomes, Pep6 could serve as a novel modulator of autophagy, potentially beneficial for cancer treatment.

## Figures and Tables

**Figure 1 ijms-25-04622-f001:**
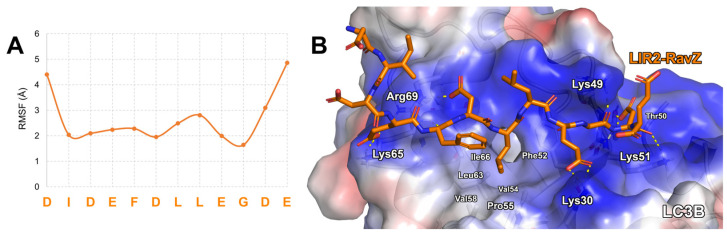
(**A**) LIR2-RavZ Cα atom RMSF plot (orange line). (**B**) Predicted binding mode of LIR2-RavZ (orange sticks) in complex with LC3B resulting at the end of MD simulations. The protein surface is colored depending on the atomic partial charges of the protein residues: blue for positive and red for negative charges, respectively. The H-bonds are represented as yellow dotted lines.

**Figure 2 ijms-25-04622-f002:**
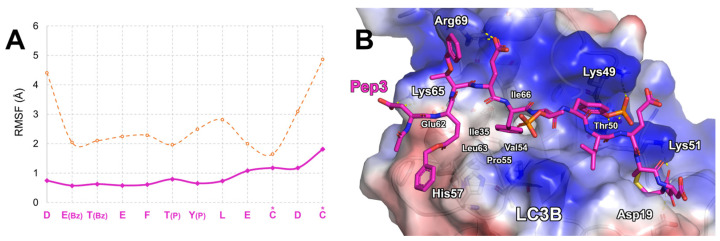
(**A**) Pep3 Cα atom RMSF plot (pink line) compared to LIR2-RavZ (orange line). Asterisks indicate the residues involved in the disulfide bond. (**B**) Predicted binding mode of Pep3 (pink sticks) in complex with LC3B resulting at the end of MD simulations. The protein surface is colored depending on the atomic partial charges of the protein residues: blue for positive and red for negative charges, respectively. The H-bonds are represented as yellow dotted lines.

**Figure 3 ijms-25-04622-f003:**
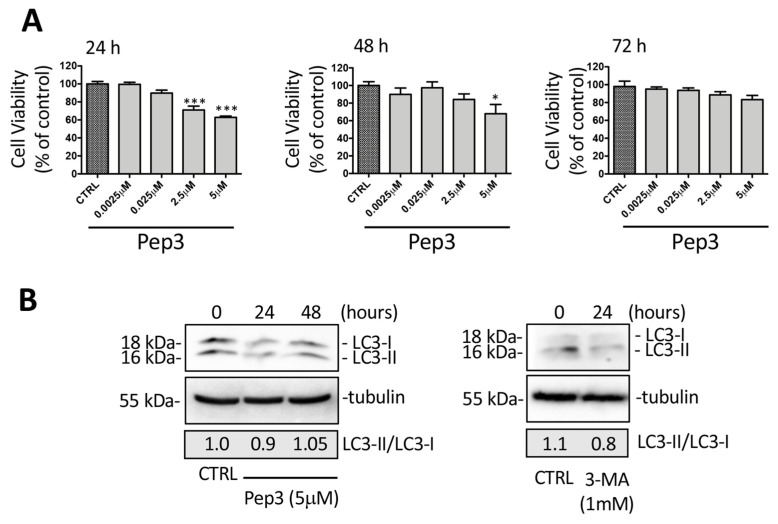
(**A**) Effect of Pep3 on PC3 cell viability. Cell viability was determined using MTT assay after 24 h, 48 h, and 72 h. Six independent biological samples for each condition were analyzed (*n* = 6). Statistical analysis was performed using one-way ANOVA followed by Dunnett’s test (* *p* < 0.05 vs. CTRL; *** *p* < 0.001 vs. CTRL). (**B**) Western blot analysis of the LC3-II/LC3-I ratio in the PC3 cells treated with Pep3 (5 µM) or 3-methyladenine (3-MA) (1 mM).

**Figure 4 ijms-25-04622-f004:**
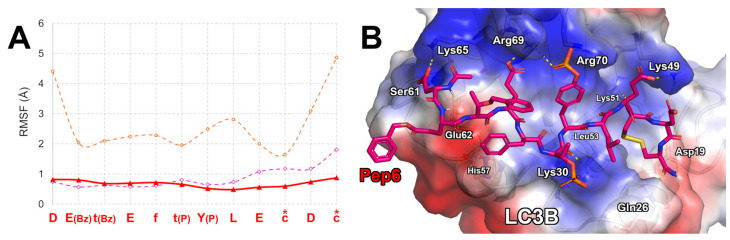
(**A**) Pep6 Cα atom RMSF plot (red line) compared to LIR2-RavZ (orange line) and Pep3 (pink line). Asterisks indicate the residues involved in the disulfide bond. The D-amino acids of the Pep6 sequence are reported as lowercase letters. (**B**) Predicted binding mode of Pep6 (magenta sticks) in complex with LC3B resulting at the end of MD simulations. The protein surface is colored depending on the atomic partial charges of the protein residues: blue for positive and red for negative charges, respectively. The H-bonds are represented as yellow dotted lines.

**Figure 5 ijms-25-04622-f005:**
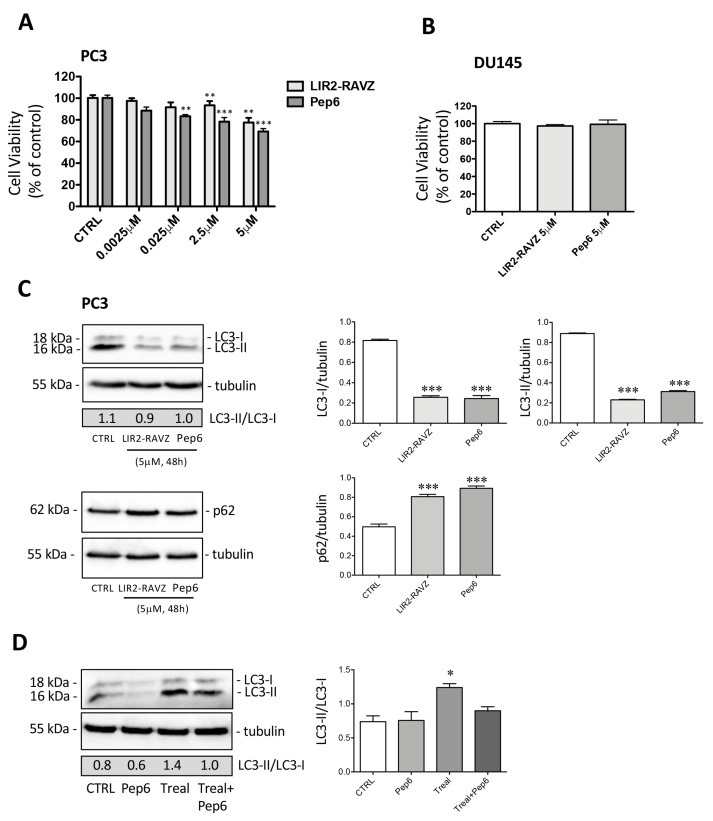
(**A**) Effect of LIR2-RavZ and Pep6 on PC3 cell viability. Cell viability was determined using MTT assay after 72 h. Six independent biological samples for each condition were analyzed (*n* = 6). Statistical analysis was performed using one-way ANOVA followed by Dunnett’s test (** *p* < 0.01 vs. CTRL; *** *p* < 0.001 vs. CTRL). (**B**) Effect of LIR2-RavZ and Pep6 on DU145 cell viability. Cell viability was determined using MTT assay after 72 h. Six independent biological samples for each condition were analyzed (*n* = 6). Statistical analysis was performed using one-way ANOVA followed by Dunnett’s test. (**C**) Western blot analysis of LC3-II/LC3-I ratio and p62 in PC3 cells treated with LIR2-RavZ and Pep6. The relative optical density of LC3-I/tubulin, LC3-II/tubulin, and p62/tubulin was quantified using ImageJ software. The bar graph represents the mean ± SD calculated from three independent experiments. Statistical analysis was performed using one-way ANOVA followed by Dunnett’s post-test (*** *p* < 0.001 vs. CTRL). (**D**) Western blot analysis of LC3-II/LC3-I ratio in PC3 cells treated with Pep6 (5 µM) and trehalose (100 mM) for 48 h. The relative optical density of LC3-II/LC3-I was quantified using ImageJ software (version 1.50i). The bar graph represents the mean ± SD calculated from three independent experiments. Statistical analysis was performed using one-way ANOVA followed by Bonferroni’s post test (* *p* < 0.05 vs. CTRL).

**Figure 6 ijms-25-04622-f006:**
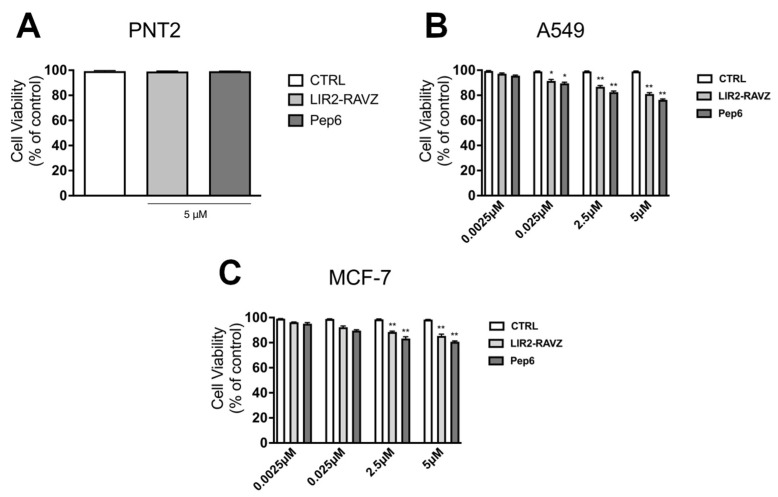
Effect of LIR2-RavZ and Pep6 on cell viability. Cell viability was determined using MTS assay on PNT2 (**A**), A549 (**B**), and MCF-7 (**C**) 96 h post-treatment. Absorbance was measured with a 96-well plate spectrophotometer (Varioskan Flash Multimode Reader) at 490 nm (* *p* < 0.05 vs. CTRL; ** *p* < 0.01 vs. CTRL).

**Figure 7 ijms-25-04622-f007:**
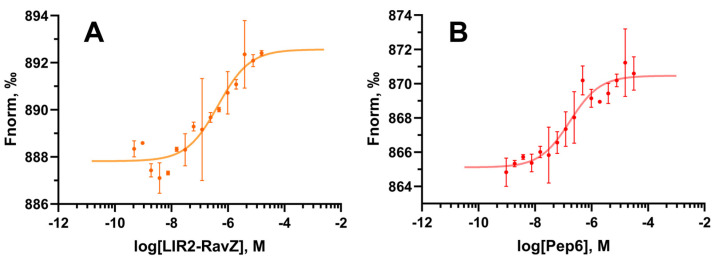
MST curves acquired using human recombinant His-tagged LC3B protein incubated with different concentrations of the control peptide LIR2-RavZ (**A**) and Pep6 (**B**), using the Monolith NT.115^Pico^ instrument. Two independent experiments were performed to compute the K_d_ curve.

**Table 1 ijms-25-04622-t001:** Primary structure and estimated binding free energy values (ΔG*) of the reference peptide LIR2-RavZ and its structural analogs.

Peptide	Sequence	ΔG* [kcal/mol]	SD
LIR2-RavZ	DIDEFDLLEGDE	−86.7	13.1
Pep1	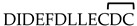	−87.5	8.7
Pep2	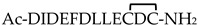	−109.1	7.2

**Table 2 ijms-25-04622-t002:** Primary structure and estimated binding free energy values (ΔG*) of the Pep2 analogs at the end of the affinity maturation procedure.

Peptide	Sequence	ΔG* [kcal/mol]	SD
Pep3	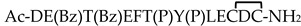	−149.8	12.1
Pep4	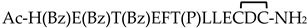	−146.5	9.5
Pep5	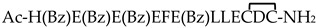	−138.2	11.6
Pep6	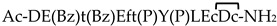	−127.1	10.8

Bn = benzyl group; P = phosphorylation on the side chain. Lowercase letters are D—amino acids.

## Data Availability

Data are contained within the article and Appendix A.

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
