# Peer review of "Computational Design of Novel Cyclic Peptides Endowed with Autophagy-Inhibiting Activity on Cancer Cell Lines"

_ijms, 2024, doi:10.3390/ijms25094622_

Round 1

Reviewer 1 Report

Comments and Suggestions for Authors

Dear all,

The authors of the manuscript describe an innovative approach to find a strong peptide binder of the Autophagy related protein LC3. This compound, termed LIR2-RavZ Cα, is further improved by computational design to afford several peptides with improved binding in silico and for one of them Pep6 in vitro. This part of the study is very sound and original. Moreover, the publication is very well written. However, the subsequent biological in cellulo evaluation of the LC3 binders lacks several critical controls to accept the publication in its current form in a high impact factor journal such as Int. J. Mol. Sci. . For the paper to be published in the current journal I would kindly ask to perform the following major and minor corrections:

Major Corrections:

-      The authors hypothesize that Lir-RavZ and its improved analogue Pep6 inhibits autophagy. However, the controls present in the manuscript are not sufficient to prove this claim. The most crucial such control would be an increase in the levels of an autophagy substrate such as the p62 protein. The decrease in the expression of LC3-I and LC-3-II as well as the increase of the LC-3-II/LC3-I ratio is not sufficient to prove that autophagy was blocked or reduced. Figure 5D is confusing the message of the paper as the LC-3-I and LC-3-II expressions as well as the ratio of LC-3-II/LC3-I are the same  in the control and in the Pep6 treated cells. This result is in direct opposition to Figure 5 C where there is a big difference between control and Pep 6 treated cells. The authors should address these conflicting results given that the same cells and incubation conditions were apparently used in both experiments.

-      The results on cell viability with Pep6 are not completely convincing as the reduction in the different cell lignes is around 20% which is not highly significant. I think that globally the cell viability results should be given less importance in the manuscript.

-      All the peptides were purchased by a company. That said there should be a proof of their structure in the supplementary information. And LC-MS analysis is indispensable to prove that such compounds were indeed produced and to assess their purity.

Minor corrections:

-The Nomenclature of the LC3 forms should be harmonized. Maybe only use pro-LC3B, LC3B-I and LC3B-II. For example, the docking was done on LC3B. What is LC3B? A consensus protein? pro-LC3B? This is not very clear in the manuscript, but it is very important for the hypothesis of the paper. If Pep6 binds to pro-LC3B than it is more logical that it inhibits that formation of LC3B-I and LC3B-II

- The authors should define better what an LC3B inhibitor means. The statement is confusing as the protein is not an enzyme. We can inhibit its function in a specific interaction. We can also inhibit a protein-protein interaction. Binder or ligand might be a more appropriate term.

- Ref 17 is quite outdated and it comes from a period where no specific autophagy inhibitors were known. The authors should better site the original publications : https://doi.org/10.1038/s41598-018-29900-x, doi 10.1080/15548627.2018.1517073,  doi: 10.4161/auto.32229

- Table 1 and Table 2: The disulfide bridges are not at the right place between the cysteines in all cases. The brackets should be adjusted between the C and C residues.

All the best,

Comments on the Quality of English Language

The publication is very well written.

Author Response

Dear all,

The authors of the manuscript describe an innovative approach to find a strong peptide binder of the Autophagy related protein LC3. This compound, termed LIR2-RavZ Cα, is further improved by computational design to afford several peptides with improved binding in silico and for one of them Pep6 in vitro. This part of the study is very sound and original. Moreover, the publication is very well written. However, the subsequent biological in cellulo evaluation of the LC3 binders lacks several critical controls to accept the publication in its current form in a high impact factor journal such as Int. J. Mol. Sci. .

>Many thanks for the positive evaluation and the appreciation of our work.

For the paper to be published in the current journal I would kindly ask to perform the following major and minor corrections:

 Major Corrections:

-      The authors hypothesize that Lir-RavZ and its improved analogue Pep6 inhibits autophagy. However, the controls present in the manuscript are not sufficient to prove this claim. The most crucial such control would be an increase in the levels of an autophagy substrate such as the p62 protein. The decrease in the expression of LC3-I and LC-3-II as well as the increase of the LC-3-II/LC3-I ratio is not sufficient to prove that autophagy was blocked or reduced.

> We agree with the Reviewer. As suggested, we analyzed p62 expression by WB after treatment with Lir-RavZ and Pep6. New Figure 5 C shows that both Lir-RavZ and Pep6 treatment increase p62 expression, confirming their ability to inhibit endogenous autophagy in PC3 cells.

- Figure 5D is confusing the message of the paper as the LC-3-I and LC-3-II expressions as well as the ratio of LC-3-II/LC3-I are the same in the control and in the Pep6 treated cells. This result is in direct opposition to Figure 5 C where there is a big difference between control and Pep 6 treated cells. The authors should address these conflicting results given that the same cells and incubation conditions were apparently used in both experiments.

> We agree with your comment and apologize for the unclear and conflicting results. The WB experiments shown in Figure 5D were conducted in the same cell line (PC3) and under the same treatment conditions of the experiments shown in Figure 5C. However, the experiments aimed at examining the combined effect of Pep6 and trehalose were carried out with a reduced cellular density and lower protein loading (10 µg of protein) compared to those in Figure 5C, to optimize conditions for this specific analysis. Under these conditions, endogenous autophagy might have been diminished, which could explain the reduced activity of Pep6 observed, even though Pep6 demonstrated significant effectiveness against trehalose-induced autophagy in previous experiments. We apologize for not reporting these differences in the Results section earlier. We have now repeated the experiments using the same cellular density and protein loading (20 µg of protein for Western blot) as in the previous experiments. The new findings are presented in Figure 5D.

-      The results on cell viability with Pep6 are not completely convincing as the reduction in the different cell lignes is around 20% which is not highly significant. I think that globally the cell viability results should be given less importance in the manuscript.

> We agree with your comment. However, we believe that these compounds could be used in inhibiting pro-survival autophagy that induces resistance to other therapies. Despite this, the cell viability results confirm the fact that these molecules do not induce cytotoxicity and that they moderately exert an anti-tumor action.

-      All the peptides were purchased by a company. That said there should be a proof of their structure in the supplementary information. And LC-MS analysis is indispensable to prove that such compounds were indeed produced and to assess their purity.

>  We apologize for this inaccuracy. In the new version of the “Supporting Material” we have added the peptides’ purity data (HPLC and Mass spectral images).

 Minor corrections:

-The Nomenclature of the LC3 forms should be harmonized. Maybe only use pro-LC3B, LC3B-I and LC3B-II. For example, the docking was done on LC3B. What is LC3B? A consensus protein? pro-LC3B? This is not very clear in the manuscript, but it is very important for the hypothesis of the paper. If Pep6 binds to pro-LC3B than it is more logical that it inhibits that formation of LC3B-I and LC3B-II

> We are grateful for the chance to clarify the computational model we used in our simulations. In several studies, for example, the one concerning the X-ray structure 5D94 with DOI: 10.1074/jbc.M115.686915, the specific state of LC3B solved is not mentioned. The structure available in the Protein Data Bank (PDB accession code 5D94) represents the sequence from amino acids 5 to 123. During the maturation process, the signal peptide spanning amino acids 121 to 125 is cleaved from the C-terminal end. Therefore, our research focused on the pro-LC3 form, as our model was developed based on this X-ray structure. Accordingly, we have added this statement to the revised manuscript.

- The authors should define better what an LC3B inhibitor means. The statement is confusing as the protein is not an enzyme. We can inhibit its function in a specific interaction. We can also inhibit a protein-protein interaction. Binder or ligand might be a more appropriate term.

> We thank you for this suggestion. We have now better specified that we are treating PPI inhibitors, since, as you stressed, LC3B is not an enzyme.

- Ref 17 is quite outdated and it comes from a period where no specific autophagy inhibitors were known. The authors should better site the original publications : https://doi.org/10.1038/s41598-018-29900-x, doi 10.1080/15548627.2018.1517073,  doi: 10.4161/auto.32229

> We thank you for this suggestion. We have now added these citations.

- Table 1 and Table 2: The disulfide bridges are not at the right place between the cysteines in all cases. The brackets should be adjusted between the C and C residues.

> We apologize for this inaccuracy. In the revised version of the manuscript we have better checked the file conversion “word to pdf”.  

Reviewer 2 Report

Comments and Suggestions for Authors

The study aimed to develop new inhibitors of the LC3B protein, which is crucial for autophagy, using an in silico drug design approach. A cyclic peptide named Pep6, with high stability, showed strong binding to LC3B and exhibited cytotoxic effects on cancer cell lines similar to a known LC3B inhibitor. Importantly, Pep6 did not affect normal cells or autophagy-defective cancer cells. These findings suggest that Pep6 could serve as a promising autophagy inhibitor for further pharmacological studies or as a basis for designing new molecules targeting autophagy.

Overall the manuscript is well written and work is well executed.

Author Response

Many thanks for the positive evaluation and the appreciation of our work.

Reviewer 3 Report

Comments and Suggestions for Authors

The paper reported computational design of novel cyclic peptides endowed with autophagy inhibiting activity. A cyclic peptide (named Pep6) was found with high conformational stability and displayed a Kd value on LC3B in the nanomolar range. Pep6 can be considered a new autophagy inhibitor that can be employed as a template for the rational design of new small molecules. Overall, the research scheme is interesting and the results obtained are good. The research deserved to be publised in this journal after minor revision, as follows:

1.      Most of the work are theoretical calculations, and the compounds used in cell experiments were obtained from outsourcer of manufacturing. However, all chemical date is missing.  In order to ensure the scientific rigor of this paper, it is imperative to include supplementary data regarding the HPLC purity and Mass spectral image for both prepared peptides. This comprehensive characterization is essential in substantiating the successful acquisition of these compounds.

2.      Some spelling errors;

Line 46:  the LC3 subfamily includes LC3A (with its two splicing variants, LC3Aα and LC3Aβ), LC3B, LC3B2, and LC3C.   ---lack of parentheses.

Line 41, Atg8, Line 286, ATG4? Line 289, ATG7 --- upper case or mixed case?

3.      All the references are labeled with two numbers.

4.      The image quality of the western blotting experiments is low, especially in the original image. It is suggested the analyze the amount of target proteins by gray scanning.

Author Response

The paper reported computational design of novel cyclic peptides endowed with autophagy inhibiting activity. A cyclic peptide (named Pep6) was found with high conformational stability and displayed a Kd value on LC3B in the nanomolar range. Pep6 can be considered a new autophagy inhibitor that can be employed as a template for the rational design of new small molecules. Overall, the research scheme is interesting and the results obtained are good.

> Many thanks for the positive evaluation and the appreciation of our work.

The research deserved to be published in this journal after minor revision, as follows:

  1. Most of the work are theoretical calculations, and the compounds used in cell experiments were obtained from outsourcer of manufacturing. However, all chemical date is missing. In order to ensure the scientific rigor of this paper, it is imperative to include supplementary data regarding the HPLC purity and Mass spectral image for both prepared peptides. This comprehensive characterization is essential in substantiating the successful acquisition of these compounds.

>  We apologize for this inaccuracy. In the new version of the Supporting material we have added the peptides purity data (HPLC and Mass spectral images).

  1. Some spelling errors; Line 46:  the LC3 subfamily includes LC3A (with its two splicing variants, LC3Aα and LC3Aβ), LC3B, LC3B2, and LC3C.   ---lack of parentheses. Line 41, Atg8, Line 286, ATG4? Line 289, ATG7 --- upper case or mixed case?

> We apologize for these typos. We have corrected both terms.

  1. All the references are labeled with two numbers.

>  We apologize for this mistake. This is an error of the insertion the manuscript in the IJMS template.

  1. The image quality of the western blotting experiments is low, especially in the original image. It is suggested the analyze the amount of target proteins by gray scanning.

> Many thanks for this comment. The quality of figures has been increased to 300 dpi in the revised manuscript.

Round 2

Reviewer 1 Report

Comments and Suggestions for Authors

Dear All,

The authors have addressed my concerns. Congratulations for performing the experiments that quickly.

All the best,